# Benefits of specialist palliative care by identifying active ingredients of service composition, structure, and delivery model: A systematic review with meta-analysis and meta-regression

**Miriam J. Johnson**[1], **Leah Rutterford**[2], **Anisha Sunny**[3], **Sophie Pask**[1], **Susanne de Wolf-Linder**[1,4], **Fliss E. M. Murtagh**[1], **Christina Ramsenthaler**[1,4¤] *

1 Wolfson Palliative Care Research Centre, Hull York Medical School, University of Hull, Hull, United Kingdom, 2 St Neots Neurological Centre, St Neots, United Kingdom, 3 School of Psychology and Social Work, Faculty of Health Sciences, University of Hull, Hull, United Kingdom, 4 School of Health Professions, Institute of Health Sciences, Zurich University of Applied Sciences, Winterthur, Switzerland

¤ Current address: Faculty of Health Professions, Institute of Nursing, Zurich University of Applied Sciences, Winterthur, Switzerland
* hycr22@hyms.ac.uk, ramn@zhaw.ch (CR)

## Abstract

### Background

Specialist palliative care (SPC) services address the needs of people with advanced illness. Meta-analyses to date have been challenged by heterogeneity in SPC service models and outcome measures and have failed to produce an overall effect. The best service models are unknown. We aimed to estimate the summary effect of SPC across settings on quality of life and emotional wellbeing and identify the optimum service delivery model.

### Methods and findings

We conducted a systematic review with meta-analysis and meta-regression. Databases (Cochrane, MEDLINE, CINAHL, ICTRP, clinicaltrials.gov) were searched (January 1, 2000; December 28, 2023), supplemented with further hand searches (i.e., conference abstracts). Two researchers independently screened identified studies. We included randomized controlled trials (RCTs) testing SPC intervention versus usual care in adults with life-limiting disease and including patient or proxy reported outcomes as primary or secondary endpoints. The meta-analysis used, to our knowledge, novel methodology to convert outcomes into minimally clinically important difference (MID) units and the number needed to treat (NNT). Bias/quality was assessed via the Cochrane Risk of Bias 2 tool and certainty of evidence was assessed using the Grading of Recommendations Assessment, Development and Evaluation (GRADE) tool. Random-effects meta-analyses and meta-regressions were used to synthesize endpoints between 2 weeks and 12 months for effect on quality of life and emotional wellbeing expressed and combined in units of MID. From 42,787 records, 39

**Data Availability Statement:** All data included were derived from publicly available documents cited in the references. All data, a data dictionary and full analysis scripts for R can be obtained on the Open Science Framework website for this project (https://osf.io/h8pmz/).

**Funding:** The University of Hull internship scheme supported LR and AS in contributing to this study. The funder of this study had no role in study design, data collection, data analysis, data interpretation, or writing of the report. FM is a UK National Institute for Health and Care Research (NIHR) Senior Investigator. The views expressed in this article are those of the author(s) and not necessarily those of the NIHR, or the Department of Health and Social Care (United Kingdom).

**Competing interests:** The authors have declared that no competing interests exist.

**Abbreviations:** CDSR, Cochrane Database of Systematic Reviews; CENTRAL, Cochrane Central Register of Controlled Trials; CI, confidence interval; CINAHL, cumulative index of nursing and allied health literature; DARE, Cochrane Database of Abstracts of Reviews of Effects; GRADE, Grading of Recommendations Assessment, Development and Evaluation; HIV, human immunodeficiency virus; HTA, Health Technology Assessment; ICTRP, International Clinical Trials Registration Plattform; MID, minimally clinically important difference; NHS EED, National Health Service Economic Evaluation Database; NNT, number needed to treat; pCG, proportion of scores in the control group; PRISMA, Preferred Reporting Items for Systematic Reviews and Meta-Analyses; QOL, quality of life; RCT, randomized controlled trial; RR, relative risk; ROB, risk of bias tool; SMD, standardized mean difference; SPC, specialist palliative; e care, UC, usual care; UK, United Kingdom; USA, United States of America.

international RCTs ($n = 38$ from high- and middle-income countries) were included. For quality of life (33 trials) and emotional wellbeing (22 trials), statistically and clinically significant benefit was seen from 3 months' follow-up for quality of life, standardized mean difference (SMD in MID units) effect size of 0.40 at 13 to 36 weeks, 95% confidence interval (CI) [0.21, 0.59], $p < 0.001$, $I^2 = 60\%$). For quality of life at 13 to 36 weeks, 13% of the SPC intervention group experienced an effect of at least 1 MID unit change (relative risk (RR) = 1.13, 95% CI [1.06, 1.20], $p < 0.001$, $I^2 = 0\%$). For emotional wellbeing, 16% experienced an effect of at least 1 MID unit change at 13 to 36 weeks (95% CI [1.08, 1.24], $p < 0.001$, $I^2 = 0\%$). For quality of life, the NNT improved from 69 to 15; for emotional wellbeing from 46 to 28, from 2 weeks and 3 months, respectively. Higher effect sizes were associated with multidisciplinary and multicomponent interventions, across settings. Sensitivity analyses using robust MID estimates showed substantial (quality of life) and moderate (emotional wellbeing) benefits, and lower number-needed-to-treat, even with shorter follow-up. As the main limitation, MID effect sizes may be biased by relying on derivation in non-palliative care samples.

## Conclusions

Using, to our knowledge, novel methods to combine different outcomes, we found clear evidence of moderate overall effect size for both quality of life and emotional wellbeing benefits from SPC, regardless of underlying condition, with multidisciplinary, multicomponent, and multi-setting models being most effective. Our data seriously challenge the current practice of referral to SPC close to death. Policy and service commissioning should drive needs-based referral at least 3 to 6 months before death as the optimal standard of care.

## Author summary

### Why was this research done?

- Specialist palliative care (SPC) services provide a complex intervention that addresses the holistic needs of individuals with life-limiting conditions and their families.

- Different intervention models include a variation of different disciplines (doctors, nurses, psychologists, physiotherapists, spiritual care workers, social care workers, etc.), configurations (e.g., whether out-of-hours care is provided), and settings (hospital, hospice, community, inpatient, outpatient, etc.).

- The overall effectiveness of SPC on quality of life and emotional wellbeing is undetermined due to large variation in intervention models and heterogeneity in outcome measures used to measure quality of life and emotional wellbeing.

### What did the researchers do and find?

- We systematically reviewed randomized controlled trials (RCTs) investigating the effectiveness of SPC, to assess which intervention model components and configurations are most effective in improving quality of life or emotional wellbeing.

- We used a method to combine effects across the range of outcome measures by converting raw scores into units of meaningful improvement.

- For the summary effects of 39 RCTs, quality of life and emotional outcomes improved from 3 months of follow-up onwards. SPC yielded a clinically meaningful effect on quality of life of moderate size. The effect was larger at 3 to 6 months than at later follow-up. For emotional wellbeing, similar effects were seen. Overall, the effect was larger for quality of life than for emotional wellbeing.

- To address the large variation in intervention models, we directly scored the number of professional groups and service elements to understand how this variation relates to outcomes. Higher effect sizes were associated with multidisciplinary and multicomponent SPC interventions, provided across healthcare settings.

### What do these findings mean?

- Our findings challenge the current practice of referring patients with life-limiting illness to SPC close to death at the end of life. Policy and service commissioning should drive needs-based referral at least 3 to 6 months before death as the optimal standard of care.

- The most effective models are multidisciplinary, multicomponent (i.e., providing more than symptom control or advance care planning) and multi-setting.

- Honoring the complex and holistic needs of patients and families by including different service elements offered by various professional groups working across settings is paramount for effective palliative care.

- Meaningful changes were not necessarily derived in specified palliative populations. Some of the effects may therefore be under- or overestimated.

### Introduction

Palliative and end-of-life care is the active, holistic care of people with advanced illness, focusing on quality of life (QoL) and symptom relief [1], applicable early in the disease course alongside disease-directed treatments [2]. Service models range from nonspecialist approaches with basic symptom management and advance care planning, to specialist services. Specialist palliative care (SPC) services address the needs of individuals and their families [3], usually delivered by a specialist multidisciplinary team (e.g., doctors, nurses, allied health professionals), fostering care coordination and collaboration between specialists and nonspecialists [3,4].

SPC team composition and service components (symptom management, rehabilitation, spiritual care, carer/bereavement support, out-of-hours services) vary dependent on healthcare settings and resources [5]. Attempts to evidence an effective SPC service delivery model are elusive [3,5–9], although various models are suggested [5], and components of service interventions identified [6]. However, a components classification is not provided, nor meta-regression conducted to estimate the summary effect on outcomes associated with specific components.

Cost-effective commissioning of palliative care resources is needed given a projected increase of 42% in the number of people requiring palliative care of all those dying by 2040 [10]. Despite 17 systematic reviews of clinical trials for SPC (see Tables A–C in S2 Appendix), these reviews and meta-analyses have been unable to produce a unanimous summary effect. The 7 meta-analyses conducted previously were often based on one quarter of eligible trials only [11–17]. They have been further hampered by major heterogeneity in outcome measures/ primary and secondary endpoints, intervention components and service models (and incomplete reporting of models) [3,8,11–17]. Most meta-analyses pool standardized mean differences (SMD) across different QoL and symptom measurement tools [11–17]; an approach challenged in the methodological literature due to issues with construct validity and scaling of different outcome measures [18,19]. The small number of included studies contributes to imprecision around the point estimate and the difficulties in interpreting the clinical relevance of findings; often the lower confidence interval bound represents questionable relevance. Only 3 reviews reported an emotional outcome measure (depression or anxiety) finding negligible (SMD 0.09) [14,15] or small (0.33) [16] effect sizes, respectively.

To address the knowledge gap and question regarding the clinical relevance of estimates of benefit, and to inform policy and clinical service commissioning, we conducted this systematic review with meta-analysis and meta-regression, to estimate the overall summary effect on QoL and emotional wellbeing of SPC across settings. We used to our knowledge novel methodology, converting raw scores into units of meaningful improvement, to combine effects across the range of outcome measures used in the original studies. Additionally, we aimed to identify the service delivery model and components associated with moderate to high effect sizes for these outcomes.

## Methods

This systematic review and meta-analysis (PROSPERO no: CRD42021292371) is reported according to the Preferred Reporting Items for Systematic Reviews and Meta-Analyses (PRISMA) guidelines (Table A in S11 Appendix) [20].

### Search strategy and selection criteria

A Boolean search strategy was used to search from January 1, 2000 to December 28, 2023, with no language restrictions (S3 Appendix). Given that SPC service models have become part of mainstream healthcare only in recent years, and that for most countries in Europe or the United States, service models and their components have been developed since the 2000s, we restricted the search to synthesizing effects for contemporary service models [21]. Database searches were supplemented with contacting field experts, hand-searching bibliographies of systematic reviews and included randomized controlled trials (RCTs), citation and reference searches, and a Web of Science search for conference abstracts. Gray literature was not searched because most unpublished or ongoing studies could be identified through the listed search strategies [22].

### Eligibility criteria and study selection

Four reviewers (LR, AS, CR, and SP) independently reviewed and selected the records against a priori eligibility criteria (eligibility criteria are summarized in Table 1). Discrepancies were resolved by consensus and arbitration by a panel of all reviewers. The Rayyan software was used to aid selection [23].

Table 1. Search strategy and eligibility criteria for study selection.

| | |
|---|---|
| Databases | • Cochrane Central Register of Controlled Trials (CENTRAL)<br>• Cochrane Database of Systematic Reviews (CDSR)<br>• Cochrane Database of Abstracts of Reviews of Effects (DARE)<br>• Health Technology Assessment (HTA)<br>• National Health Service Economic Evaluation Database (NHS EED) (all via Cochrane Library at Wiley Interscience<br>• MEDLINE & MEDLINE In-Process (via Ovid and PubMed, from 1947)<br>• CINAHL (via Ebscohost, from 1982)<br>• International Clinical Trials Registry Platform<br>• US National platform clinicaltrials.gov |
| Abbreviated search strategy | We used the search terms ("End of life" OR "end stage" OR "advanced care plan*" OR hospice OR terminal* OR dying OR incurable OR palliat*) AND (intervention* OR care OR therap* OR support* OR service*) AND randomi*, using both keywords and Medical Subject Heading terms (see S3 Appendix). |
| Design | • Single-blinded or non-blinded parallel randomized controlled trials (RCTs).<br>• At least 1 follow-up time point.<br>• Phase III RCTs or Phase II (feasibility studies) were considered eligible if they included a clear efficacy endpoint and presented an a priori sample size calculation to ensure adequate power.<br>• Cluster RCTs were included with the appropriate adjustments [24].<br>• The primary endpoint was at 3 months post intervention with secondary endpoints included from 2 weeks to a maximum of 12 months post intervention [11–17]. |
| Population | • Adults (18+) with advanced illness with palliative care needs.<br>• Study samples could comprise homogeneous groups defined by a specific advanced illness or could be comprised of diverse conditions with clear palliative care needs.<br>• Studies of samples with a primary presentation of HIV due to medical advances in the treatment of this condition were excluded. |
| Intervention | • SPC interventions that were multicomponent and delivered by a multidisciplinary team in any setting, comprising more than one of the core elements of palliative care [3,6].<br>• We excluded all nonspecialist palliative care interventions, single-component interventions consisting solely of advance care planning or interventions not delivering direct patient care (encompassing professional education or training programs, and family and caregiver-oriented interventions).<br>• Nonspecialist palliative care was excluded due to the cross-contamination of methods resulting from the influences of SPC personnel and research. |
| Comparator | • Usual care, defined as care provided by personnel that are not designated SPC professionals, without SPC input (e.g., standard oncological care alone) in any setting [11], at the point of entry to the RCT. |
| Outcomes | • All patient-centered multidimensional health-related QoL, QoL, symptom or psychosocial measures addressing the core components of palliative care.<br>• Demonstrated validity and reliability.<br>• As the primary aim was to explore the relationship between the components of care and effectiveness in relation to QoL or emotional wellbeing of the patient, all other outcomes (function, quality of care, health service utilization, mortality, costs, sole family, or caregiver outcomes) were excluded. |

HIV, human immunodeficiency virus; RCT, randomized controlled trial; SPC, specialist palliative care.

## Data extraction and risk of bias assessment

A data extraction spreadsheet was developed (CR) and piloted within the team over 10% of included papers. Two reviewers independently extracted the data (AS) and outcomes (LR), with each verifying the other's work. Where studies missed information or provided minimal details, study authors or protocols/further analyses were reviewed. Any disagreements were resolved via a third author (MJJ/CR).

Two reviewers independently assessed risk of bias using the Cochrane risk of bias assessment tool (RoB2) (LR, CR) [25]. We scored the certainty of evidence with the Grading of

Recommendations Assessment, Development and Evaluation (GRADE) framework (Table A and Fig A in S8 Appendix) [26,27].

## Classification of components of the specialist palliative care intervention model

For the meta-regression, the number of components of SPC was used as one regressor to explain between-study variation in effect sizes. We developed a classification scheme for scoring essential components of SPC services based on prior reviews (Tables A and B in S1 Appendix) [5–9,28].

## Data analysis

Data were extracted as presented in the reports for all endpoints for our primary outcomes of QoL and emotional wellbeing, according to the scoring manual of each measure. If a study presented data from several eligible outcome measures, the outcome (i) most commonly used in the included studies; and (ii) for which an anchor-based or distribution-based minimal important difference (MID) had been derived, was chosen as we meta-analyzed SMD expressed in MID units as the summary effect (see Tables A and B in S5 Appendix for a list of all MIDs used) [18]. The direction of scales was converted so that a positive MID shows a benefit for the SPC group. Similar to other meta-analysis [11–17], we extracted data for the following time points: 2 to 11 weeks, 12, 13 to 36 weeks, and 37 weeks to 12 months follow-up. If a study reported several time points within these time windows, we took the point closest to the upper bound of each interval. In multi-arm trials, only data on arms meeting our eligibility criteria were extracted.

Missing information was imputed using the conversion formulae reported in the Cochrane handbook [29]. Data from cluster RCTs was adjusted by the reported or imputed intraclass correlation coefficient, as per Cochrane methodology [29]. We performed each random-effects meta-analysis on the basis of MID units (full methodology: S4 Appendix) [18,19], addressing the limitations of the traditional SMD when pooling different outcome measures and baseline variability in the population. An SMD(MID) of 0.50 or higher indicates substantial benefit [19].

For each outcome and endpoint, random-effects meta-analyses using the Hartung–Knapp correction pooled results (i) in MID units, performing an inverse variance weighted, random-effects meta-analysis; and (ii) as relative risks (RR) (quotient of probability of experiencing >1 MID change in the SPC group versus control group). From this RR, the number needed to treat (NNT) was derived via the formula given in S4 Appendix as: 1 divided by the probability of experiencing >1 MID change in the control group times 1 minus the RR [18]. Here, the NNT describes the number of patients that need to be treated with SPC in order for at least 1 patient to experience a benefit of change in the outcome of at least 1 MID unit. We used the Cochran $X^2$ test, $I^2$ and the $\tau$ statistic to evaluate statistical heterogeneity (0% to 40% small, 30% to 60% moderate, 50% to 90% substantial, >75% considerable) [29]. Univariate linear meta-regressions were used to understand the variability in MID effect size across studies. The main covariate investigated was the service composition score, derived from the adapted classification system (see S1 Appendix). Additional covariates in univariate meta-regression analyses were year of publication, RoB2 quality score (low/some/high), % attrition, population (cancer/non-cancer/mixed), and setting (inpatient consulting model/home or hospital outreach/multiple settings).

Publication bias was assessed using enhanced funnel plots and the Egger test. All statistical analyses were performed using R v4.0.1. All comparisons were two-tailed using a threshold $p < 0.05$.

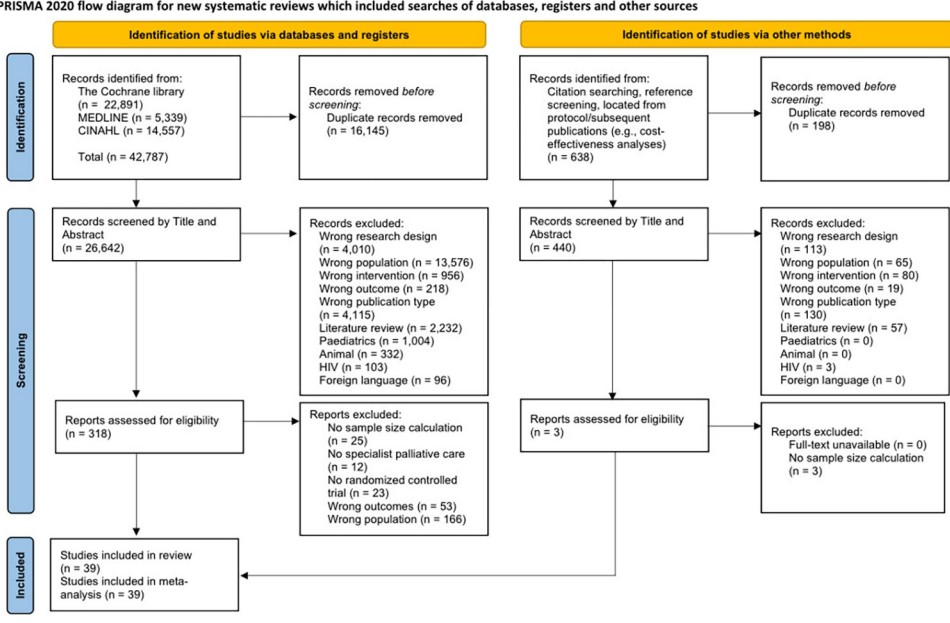

**Fig 1. PRISMA flow diagram.**

## Results

The database search yielded 42,787 records (Fig 1), with 638 references located through other sources. Following deduplication, 26,642 titles and abstracts were screened, of which 39 RCTs (6,089 patients in total: 3,023 SPC and 3,066 usual care) were included (reasons for exclusion, see S6 Appendix). All studies were published between 2002 and 2022. In terms of countries, 16 RCTs were done in the United States of America, 4 in Italy, 3 in Denmark or the United Kingdom, 2 each in Australia or China, and 1 each in Belgium, Brazil, Canada, the Czech Republic, India, South Korea, Sweden, Switzerland, and the Netherlands.

Twenty single and 19 multi-center studies provided hospital-based inpatient, outpatient, hospice, and community-based SPC interventions (Table B in S7 Appendix). One study used an inpatient ward-based SPC model [30], 5 an inpatient consulting model [31–35], 12 a hospital outpatient model [36–47], 2 a hospital outreach model [48,49], 2 a community-based SPC intervention [50,51], and the remaining 17 RCTs had SPC interventions spanning multiple settings [52–68]. Table 2 specifies the number of team members, the SPC services provided, and team availability. Nine RCTs only included 1 profession, either physicians or nurses. No study included all professions. RCTs by Liu and colleagues [30] and Nottelmann and colleagues [41] included the most diverse group of professions. All SPC interventions in included RCTs used symptom assessment and management, also of psychosocial symptoms and wider unmet needs. Advance care planning was part of 62% (24/39) of RCTs, 79% (31/39) provided carer support prior to the patient's death. Only 4 studies included physiotherapy/rehabilitation services [30,36,41,48], 2 offered bereavement support [30,58]. All SPC interventions included an initial assessment, with 87% (34/39) offering planned follow-up and 41% (16/39) patient-initiated follow-up. Twelve of 39 RCTs (31%) offered an initial assessment, scheduled and patient-initiated follow-up. Only 1 study provided out-of-hours availability [50].

Most (64%, 25/39 RCTs) tested SPC versus usual care in populations with cancer (Table A in S7 Appendix). Of these, 19 RCTs used populations with different cancer diagnoses. Fourteen studies had populations with different non-cancer conditions, in heart failure or mixed

**Table 2. The model of SPC table; service composition scores and weights given to the SPC intervention description per individual trial.**

| Study | Physician | Nurse | Psychologist | Physiotherapist | Occupational health | Social worker | Spiritual support | Volunteers | Subtotal | Symptom management | Advance care planning | Carer support | Bereavement care | Rehabilitation | Education/liaison | Subtotal | Initial assessment | Planned follow-up | Patient-initiated follow-up | Out-of-hours availability | Subtotal | Total |
|---|---|---|---|---|---|---|---|---|---|---|---|---|---|---|---|---|---|---|---|---|---|---|
| | | | | Multidisciplinary team | | | | | | | | Services provided | | | | | | | Availability | | | |
| Subscore | 3 | 3 | 1 | 2 | 2 | 2 | 2 | 1 | 16 | 1 | 1 | 1 | 1 | 1 | 1 | 6 | 1 | 1 | 1 | 1 | 4 | 26 |
| Aiken et al 2006 [52] | * | * | | | | * | * | | 10 | * | * | * | | | * | 4 | * | * | | | 2 | 16 |
| Bakitas et al 2009 [53] | | * | | | | | | | 3 | * | * | * | | | * | 4 | * | * | * | | 3 | 10 |
| Bakitas et al 2015 [54] | | * | | | | | | | 3 | * | * | * | | | * | 4 | * | * | * | | 3 | 10 |
| Bakitas et al 2020 [55] | | * | | | | | | | 3 | * | * | * | | | * | 4 | * | * | * | | 3 | 10 |
| Bassi et al 2021 [36] | * | * | * | * | | | | | 9 | * | * | * | | * | | 4 | * | * | | | 2 | 15 |
| Bekelman et al 2018 [56] | * | * | | | | * | | | 8 | * | | * | | | * | 3 | * | * | * | | 3 | 14 |
| Bekelman et al 2022 [37] | * | * | | | | * | | | 8 | * | | * | | | * | 3 | * | * | * | | 3 | 14 |
| Benthien et al 2020 [57] | * | * | * | | | | | | 7 | * | | * | | | * | 3 | * | * | | | 2 | 12 |
| Brännström et al 2014 [48] | * | * | | * | * | | | | 10 | * | | | | * | * | 3 | * | * | | | 2 | 15 |
| Brims et al 2019 [49] | * | | | | | | | | 3 | * | | * | | | * | 3 | * | * | | | 2 | 8 |
| Do Carmo et al 2017 [31] | * | | | | | | | | 3 | * | | | | | * | 2 | * | * | | | 2 | 7 |
| Edmonds et al 2010 [58] | * | * | | | | * | | | 8 | * | * | * | * | | * | 5 | * | * | | | 2 | 15 |
| El-Jahwari et al 2016 [32] | * | * | | | | | | | 6 | * | * | | | | | 2 | * | * | * | | 3 | 11 |
| El-Jahwari et al 2021 [33] | * | * | | | | | | | 6 | * | * | | | | * | 3 | * | * | * | | 3 | 12 |
| Evans et al 2021 [59] | | * | | | | | | | 3 | * | * | * | | | * | 4 | * | * | * | | 3 | 10 |
| Eychmüller et al 2021 [38] | * | * | | | | | | | 6 | * | * | | | | * | 3 | | * | | | 1 | 10 |
| Franciosi et al 2019 [60] | * | * | | | | | | | 6 | * | * | * | | | * | 4 | * | * | | | 2 | 12 |
| Gao et al 2020 [61] | * | * | | | | | | | 6 | * | * | * | | | * | 4 | * | * | | | 2 | 12 |
| Given et al 2002 [62] | | | * | | | * | | | 3 | * | | * | | | | 2 | * | * | | | 2 | 7 |
| Goldstein et al 2022 [50] | * | * | | | | * | | | 8 | * | * | | | | * | 3 | * | * | | * | 3 | 14 |
| Greer et al 2022 [34] | * | * | | | | | | | 6 | * | * | | | | * | 3 | * | * | | | 2 | 11 |
| Groenvold et al 2017 [63] | * | * | | | | | | | 6 | * | | | | * | | 2 | | * | * | | 2 | 10 |
| Hoek et al 2017 [64] | * | * | | | | | | | 6 | * | | | | | * | 2 | * | * | | | 2 | 10 |
| Kluger et al 2020 [39] | * | * | | | | * | * | | 10 | * | * | | | * | * | 4 | * | * | | | 2 | 16 |
| Liu et al 2022 [30] | * | * | * | * | *1 | * | * | | 14 | * | * | * | * | * | * | 6 | | * | * | | 2 | 22 |
| Maltoni et al 2016 [40] | * | * | | | | | | | 6 | * | * | | | * | * | 4 | * | * | | | 2 | 12 |
| Nottelmann et al 2021 [41] | * | * | * | * | * | * | * | | 15 | * | * | * | | * | * | 5 | * | * | | | 2 | 22 |
| Patil et al 2021 [65] | * | * | * | | | * | * | | 11 | * | | * | | | | 2 | * | * | * | | 3 | 16 |
| Rogers et al 2017 [66] | * | * | | | | | | | 6 | * | * | * | | | * | 4 | * | * | * | | 3 | 13 |

*(Continued)*

**Table 2.** (Continued)

| | Multidisciplinary team | | | | | | | Services provided | | | | | | | | Availability | | | | |
|---|---|---|---|---|---|---|---|---|---|---|---|---|---|---|---|---|---|---|---|---|
| Scarpi et al 2019 [42] | * | * | | | | | 6 | * | * | * | | * | 4 | * | * | * | | 2 | 12 |
| Sidebottom et al 2015 [35] | * | * | | | | | 6 | * | * | * | | * | 4 | * | * | | * | 2 | 12 |
| Slama et al 2020 [43] | * | | | | | | 3 | * | * | | | * | 2 | * | * | | | 2 | 7 |
| Tattersall et al 2014 [44] | | * | | | | | 3 | * | * | | | | 1 | * | | * | | 2 | 6 |
| Temel et al 2010 [45] | * | * | | | | | 6 | * | * | * | | * | 4 | * | * | | * | 2 | 12 |
| Temel et al 2020 [67] | * | * | | | | | 6 | * | * | * | | * | 4 | * | * | * | | 2 | 12 |
| Vanbutsele et al 2020 [68] | * | * | * | | | | 7 | * | * | * | | * | 4 | * | * | * | * | 3 | 14 |
| Wong et al 2016 [51] | * | * | | | * | | 7 | * | * | * | | * | 4 | * | * | * | | 2 | 13 |
| Woo et al 2019 [46] | | * | * | | | | 4 | | | | | | 2 | * | * | | | 2 | 8 |
| Zimmermann et al 2014 [47] | * | * | | | | | 6 | * | * | * | | * | 4 | * | * | * | * | 3 | 13 |

¹Liu and colleagues [30] also had a nutritionist on the multidisciplinary team, scored with 1 point as the assessment was not offered to all trial arm patients.

conditions. The mean age was 65.8 years, 47.3% female, a median of 14% of non-white patients included in individual studies, and 55% of patients across RCTs had advanced illness. The median proportion with a low performance status was 35% (see Table D in S7 Appendix).

Despite the inclusion criterion of RCTs with a priori sample size calculation, 19 included studies were scored as "some risk," and 14 studies as "high risk" of bias (see S8 Appendix). Attrition in high-risk studies ranged from 2% to 61% with a median of 24%. Non-blinding and attrition lead GRADE downgrading due to wide confidence intervals.

All included RCTs measured QoL as either a primary or secondary outcome, using various generic and disease-specific outcome measures (see Table C in S7 Appendix). After converting data into MID units, we pooled results for our primary endpoint at 13 to 36 weeks follow-up. A moderate to substantial effect of SMD(MID) 0.40 (95% CI [0.21, 0.59], $p < 0.001$, $I^2 = 60\%$) was seen, based on 33 studies with $n = 4,493$ (SPC: 2,219, usual care (UC): 2,274). This converts into a relative risk of $RR = 1.13$ (95% CI [1.06, 1.20], $p < 0.001$, $I^2 = 0\%$); at least 13% of the SPC intervention group experience an effect of at least 1 MID unit. This proportion benefiting from the SPC intervention by at least 1 MID change increased to 31% at 7 months to 1-year post-intervention, although this effect relied on a small number of studies ($k = 9$). The NNT for the primary endpoint at 13 to 36 weeks was 20; 20 patients need to be treated for one patient to experience a 1 MID change in the QoL outcome. The NNT for the 7 to 12 months follow-up was 12. Table 3 shows the results of meta-regression analyses for all 3 meta-analyses of the QoL outcome at different time points, and Fig 2 presents the forest plot for the primary endpoint with the relative risk as the effect size (see also S9 Appendix for a complete documentation of all results).

The covariate consistently associated with the SMD(MID) was the attrition rate with studies yielding lower effect sizes with higher attrition rates. For the endpoint 13 to 36 weeks follow-up, the total service composition score showed a moderate, statistically significant association with the effect size ($\beta = 0.07$, 95% CI [0.03, 0.11], $k = 33$ studies, $n = 4,493$). Publication bias (Egger's test) was only present for the main endpoint (see S9 Appendix).

Two sensitivity analyses with a different grouping of endpoints and using the most robust MID for the meta-analyses yielded substantial MID effect sizes of SMD(MID) 0.66 at 12 to 16 weeks ($p < 0.001$) and 0.53 at 13 to 36 weeks ($p = 0.003$) with NNTs of 14 and 13, respectively. The service composition score was no longer a statistically significant covariate (S10 Appendix).

A higher homogeneity in outcome measures was observed for emotional wellbeing, with the majority (8/39) of studies measuring anxiety and depression with the Hospital Anxiety and Depression Scale. However, fewer RCTs evaluated emotional wellbeing, reflected in the less precise confidence intervals despite comparable NNTs to the QoL outcome, although relative risk values showed statistically significant benefit at our primary endpoint of 13 to 36 weeks (see Table 3 and Fig 3). For emotional wellbeing, the RR effect sizes at the endpoint 12 weeks (RR 1.12, 95% CI [1.01, 1.25], $k = 18$, $n = 2,266$, $I^2 = 0\%$) and at 13 to 36 weeks (RR 1.16, 95% CI [1.08, 1.24], $k = 22$, $n = 2,728$, $I^2 = 0\%$) yielded statistically significant results with 12% to 16% of the SPC intervention group experiencing at least a 1 MID change and resulting in an NNT of 19 to 28.

The service composition score emerged as consistently associated with the effect size. The effect of a higher service composition score showed moderate to large positive $\beta$ coefficients; a higher complexity of service was associated with larger MID effect sizes. The moderating effect of service composition and complexity was largest at 2 weeks to 3 months follow-up for the emotional wellbeing outcome with $\beta = 0.29$, 95% CI [0.16, 0.43], $k = 14$ studies, $n = 2,196$.

Two sensitivity analysis with robust MID estimates yielded a moderate effect with $RR = 1.23$ (95% CI [1.08, 1.39], $p = 0\cdot002$, $I^2 = 46\%$) at 12 to 16 weeks follow-up. The NNT was

**Table 3. Overview of meta-analysis results according to outcome and endpoint.**

| Analysis | k | SMD (MID) [95% CI] | p | $I^2$ | RR (95% CI) | p | $I^2$ | Interpretation | NNT[1] | Covariates (meta-regression)[2] |
|---|---|---|---|---|---|---|---|---|---|---|
| *Quality of life outcome* | | | | | | | | | | |
| #1 Quality of life at 2 to 11 weeks | 20 | 0.16 [-0.06, 0.38] | 0.136 | 50% | 1.04 [0.95; 1.14] | 0.384 | 0% | 4% in SPC group have benefit of at least 1 MID change | 69 | Attrition Quality score (low) |
| #2 Quality of life at 12 weeks | 27 | 0.50 [0.06, 0.93] | **0.028** | 97% | 1.14 [0.95; 1.36] | 0.149 | 44% | 14% in SPC group have benefit of at least 1 MID change | 15 | Attrition (Non-cancer)[3] |
| #3 Quality of life at 13 to 36 weeks | 33 | 0.40 [0.21, 0.59] | **<0.001** | 60% | 1.13 [1.06; 1.20] | **<0.001** | 0% | 13% in SPC group have benefit of at least 1 MID change | 20 | Attrition (Non-cancer) Quality score (low) Model of SPC (higher score) Multiple settings (vs. Single settings) |
| #4 Quality of life at 37 weeks to 12 months | 9 | 0.58 [-0.09, 1.26] | 0.079 | 74% | 1.31 [1.08; 1.59] | **0.012** | 0% | 31% in SPC group have benefit of at least 1 MID change | 12 | Attrition Home or hospital outreach/ multiple settings |
| **Sensitivity analyses** | | | | | | | | | | |
| Quality of life at 12 to 16 weeks with most robust MID | 34 | 0.66 [0.35, 0.98] | **<0.001** | 96% | 1.20 [1.07; 1.35] | **0.004** | 32% | 20% in SPC group have benefit of at least 1 MID change | 14 | Attrition Quality score (low) |
| Quality of life at 13 to 36 weeks with most robust MID | 35 | 0.53 [0.20, 0.87] | **0.003** | 96% | 1.22 [1.06; 1.40] | **0.008** | 44% | 22% in SPC group have benefit of at least 1 MID change | 13 | Attrition Inpatient SPC model |
| *Emotional wellbeing outcome* | | | | | | | | | | |
| #5 Emotional wellbeing at 2 to 11 weeks | 14 | 0.18 [−0.64, 0.99] | 0.643 | 92% | 1.07 [0.85, 1.35] | 0.517 | 69% | 7% in SPC group have benefit of at least 1 MID change | 46 | (Attrition) Model of SPC (higher score) Quality score (low) |
| #6 Emotional wellbeing at 12 weeks | 18 | 0.08 [-0.06, 0.23] | 0.254 | 49% | 1.12 [1.01, 1.25] | **0.039** | 0% | 12% in SPC group have benefit of at least 1 MID change | 28 | (Attrition) Model of SPC (higher score) Home or hospital outreach |
| #7 Emotional wellbeing at 13 to 36 weeks | 22 | 0.26 [−0.00, 0.52] | 0.053 | 82% | 1.16 [1.08, 1.24] | **<0.001** | 0% | 16% in SPC group have benefit of at least 1 MID change | 19 | Model of SPC Integrated collaborative care |

(*Continued*)

**Table 3.** (Continued)

| Analysis | k | SMD (MID) [95% CI] | p | I² | RR (95% CI) | p | I² | Interpretation | NNT[1] | Covariates (meta-regression)[2] |
|---|---|---|---|---|---|---|---|---|---|---|
| #8 Emotional wellbeing at 37 weeks to 12 months | 7 | 0.10 [−0.21, 0.41] | 0.461 | 38% | 0.97 [0.84, 1.12] | 0.593 | 0% | No benefit in SPC group of at least 1 MID change | - | - |
| **Sensitivity analyses** | | | | | | | | | | |
| Emotional wellbeing at 12 to 16 weeks with most robust MID | 23 | 0.56 [0.04, 0.98] | **0.033** | 88% | 1.23 [1.08, 1.39] | **0.002** | 46% | 22% in SPC group have benefit of at least 1 MID change | 14 | Attrition Model of SPC (higher score) Home or hospital outreach |
| Emotional wellbeing at 13 to 36 weeks with most robust MID | 22 | 0.12 [−0.06, 0.29] | 0.176 | 78% | 1.13 [1.05, 1.22] | **0.003** | 0% | 13% in SPC group have benefit of at least 1 MID change | 25 | Attrition Model of SPC (higher score) |

[1]NNT: The number needed to treat was determined by the formula presented in Thorlund and colleagues and is based on the relative risk via $1 / (p_{CG} * (1 - \text{relative risk}))$ with $p_{CG}$ being the proportion of participants in the control group experiencing a benefit of at least 1 MID change (number of participants with at least 1 MID change divided by the total number of participants in the control group).

[2]Covariates were entered into univariable univariate mixed-effects meta-regression analyses. Statistical significance was determined for the beta regression coefficients within a random effects model.

[3]Bracketed covariates have p-values of $p < 0.10$.

CI, confidence interval; k, number of included trials; I², % of heterogeneity; MID, minimal important difference; Model of SPC, Model score of specialist palliative care components (see Table 2); NNT, number needed to treat; p, p-value; RR, relative risk; SMD(MID), standardized mean difference expressed in minimal important difference units; SPC, specialist palliative care.

14. The positive effect stemmed from 5 studies [28–31,65], all measuring anxiety and showing benefit in mixed cancer or populations with hematological cancers. An NNT of 25 was obtained for the 13 to 36 weeks time point using the most robust MID (k = 22 studies; S10 Appendix).

## Discussion

Specialist palliative care, provided in addition to usual care, provided clinically important improvements in QoL and emotional wellbeing, with moderate effect size. For QoL and emotional outcomes, statistically and clinically significant benefit was seen in trials with follow-up for 3 months or more. The NNT for 1 patient to have benefit of at least 1 MID improved for both types of measures in studies with follow-up of between 3 and 6 months. The NNT for QoL outcomes improved from 69 (<3 months) to 15 (12 weeks) and 20 (13+ weeks), and the NNT for emotional outcomes improved from 46 (<3 months) to 19 (13 to 36 weeks). Greater effect sizes were seen in trials of higher quality; with lower attrition; in non-cancer study populations; and with multidisciplinary palliative care interventions providing a range of components (e.g., symptom control, psycho-spiritual care, support for family carers, etc.); and in provision across healthcare settings (hospital, hospice, and community). In the sensitivity analysis, statistically and clinically significant benefit was seen for both types of outcomes even in studies with follow-up of <3 months.

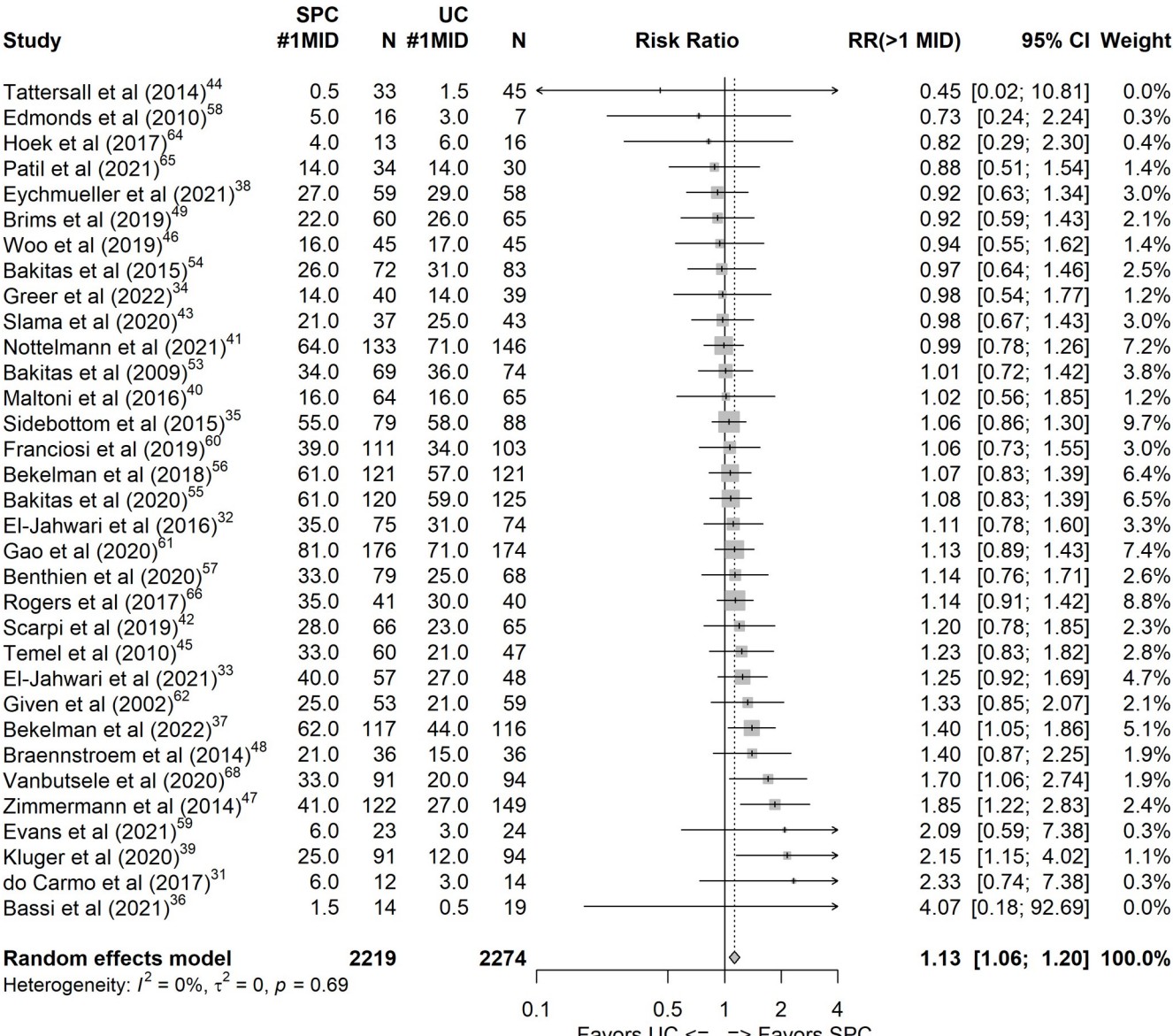

| Study | SPC #1MID | N | UC #1MID | N | Risk Ratio | RR(>1 MID) | 95% CI | Weight |
|---|---|---|---|---|---|---|---|---|
| Tattersall et al (2014)[44] | 0.5 | 33 | 1.5 | 45 | | 0.45 | [0.02; 10.81] | 0.0% |
| Edmonds et al (2010)[58] | 5.0 | 16 | 3.0 | 7 | | 0.73 | [0.24; 2.24] | 0.3% |
| Hoek et al (2017)[64] | 4.0 | 13 | 6.0 | 16 | | 0.82 | [0.29; 2.30] | 0.4% |
| Patil et al (2021)[65] | 14.0 | 34 | 14.0 | 30 | | 0.88 | [0.51; 1.54] | 1.4% |
| Eychmueller et al (2021)[38] | 27.0 | 59 | 29.0 | 58 | | 0.92 | [0.63; 1.34] | 3.0% |
| Brims et al (2019)[49] | 22.0 | 60 | 26.0 | 65 | | 0.92 | [0.59; 1.43] | 2.1% |
| Woo et al (2019)[46] | 16.0 | 45 | 17.0 | 45 | | 0.94 | [0.55; 1.62] | 1.4% |
| Bakitas et al (2015)[54] | 26.0 | 72 | 31.0 | 83 | | 0.97 | [0.64; 1.46] | 2.5% |
| Greer et al (2022)[34] | 14.0 | 40 | 14.0 | 39 | | 0.98 | [0.54; 1.77] | 1.2% |
| Slama et al (2020)[43] | 21.0 | 37 | 25.0 | 43 | | 0.98 | [0.67; 1.43] | 3.0% |
| Nottelmann et al (2021)[41] | 64.0 | 133 | 71.0 | 146 | | 0.99 | [0.78; 1.26] | 7.2% |
| Bakitas et al (2009)[53] | 34.0 | 69 | 36.0 | 74 | | 1.01 | [0.72; 1.42] | 3.8% |
| Maltoni et al (2016)[40] | 16.0 | 64 | 16.0 | 65 | | 1.02 | [0.56; 1.85] | 1.2% |
| Sidebottom et al (2015)[35] | 55.0 | 79 | 58.0 | 88 | | 1.06 | [0.86; 1.30] | 9.7% |
| Franciosi et al (2019)[60] | 39.0 | 111 | 34.0 | 103 | | 1.06 | [0.73; 1.55] | 3.0% |
| Bekelman et al (2018)[56] | 61.0 | 121 | 57.0 | 121 | | 1.07 | [0.83; 1.39] | 6.4% |
| Bakitas et al (2020)[55] | 61.0 | 120 | 59.0 | 125 | | 1.08 | [0.83; 1.39] | 6.5% |
| El-Jahwari et al (2016)[32] | 35.0 | 75 | 31.0 | 74 | | 1.11 | [0.78; 1.60] | 3.3% |
| Gao et al (2020)[61] | 81.0 | 176 | 71.0 | 174 | | 1.13 | [0.89; 1.43] | 7.4% |
| Benthien et al (2020)[57] | 33.0 | 79 | 25.0 | 68 | | 1.14 | [0.76; 1.71] | 2.6% |
| Rogers et al (2017)[66] | 35.0 | 41 | 30.0 | 40 | | 1.14 | [0.91; 1.42] | 8.8% |
| Scarpi et al (2019)[42] | 28.0 | 66 | 23.0 | 65 | | 1.20 | [0.78; 1.85] | 2.3% |
| Temel et al (2010)[45] | 33.0 | 60 | 21.0 | 47 | | 1.23 | [0.83; 1.82] | 2.8% |
| El-Jahwari et al (2021)[33] | 40.0 | 57 | 27.0 | 48 | | 1.25 | [0.92; 1.69] | 4.7% |
| Given et al (2002)[62] | 25.0 | 53 | 21.0 | 59 | | 1.33 | [0.85; 2.07] | 2.1% |
| Bekelman et al (2022)[37] | 62.0 | 117 | 44.0 | 116 | | 1.40 | [1.05; 1.86] | 5.1% |
| Braennstroem et al (2014)[48] | 21.0 | 36 | 15.0 | 36 | | 1.40 | [0.87; 2.25] | 1.9% |
| Vanbutsele et al (2020)[68] | 33.0 | 91 | 20.0 | 94 | | 1.70 | [1.06; 2.74] | 1.9% |
| Zimmermann et al (2014)[47] | 41.0 | 122 | 27.0 | 149 | | 1.85 | [1.22; 2.83] | 2.4% |
| Evans et al (2021)[59] | 6.0 | 23 | 3.0 | 24 | | 2.09 | [0.59; 7.38] | 0.3% |
| Kluger et al (2020)[39] | 25.0 | 91 | 12.0 | 94 | | 2.15 | [1.15; 4.02] | 1.1% |
| do Carmo et al (2017)[31] | 6.0 | 12 | 3.0 | 14 | | 2.33 | [0.74; 7.38] | 0.3% |
| Bassi et al (2021)[36] | 1.5 | 14 | 0.5 | 19 | | 4.07 | [0.18; 92.69] | 0.0% |
| **Random effects model** | | **2219** | | **2274** | | **1.13** | **[1.06; 1.20]** | **100.0%** |

Heterogeneity: $I^2 = 0\%$, $\tau^2 = 0$, $p = 0.69$

0.1   0.5  1  2  4
Favors UC <= => Favors SPC

**Fig 2. Forest plot of the 13 to 36 months endpoint for the quality of life outcome, effect size: relative risk of experiencing a change of $\geq$ 1 MID between baseline and the endpoint.** #1MID, number of participants in the respective group experiencing change of at least 1 minimal important difference; CI, confidence interval; MID, minimal important difference; *N*, total number of participants in group; RR, relative risk; SPC, specialist palliative care; UC, usual care.

We were able to demonstrate a moderate effect size for clinically meaningful benefit, in contrast to the uncertainty in previous published evidence syntheses [3,5–9,11–17], identify the timescale of expected benefit, and present benefit in a clinically relevant format (the NNT).

To contextualize the relevance of these results, it is important to understand that currently, SPC services are most accessed within the last days and weeks of life. A systematic review examining SPC duration (169 studies, 23 countries, 11,996,479 patients) found a median duration before death of only 18.9 days (interquartile range 0.1); shorter for people with non-cancer diseases (15 days cancer versus 6 days non-cancer) [69]. Even in countries with well-established SPC with national coverage, most referrals are made less than 3 months before death

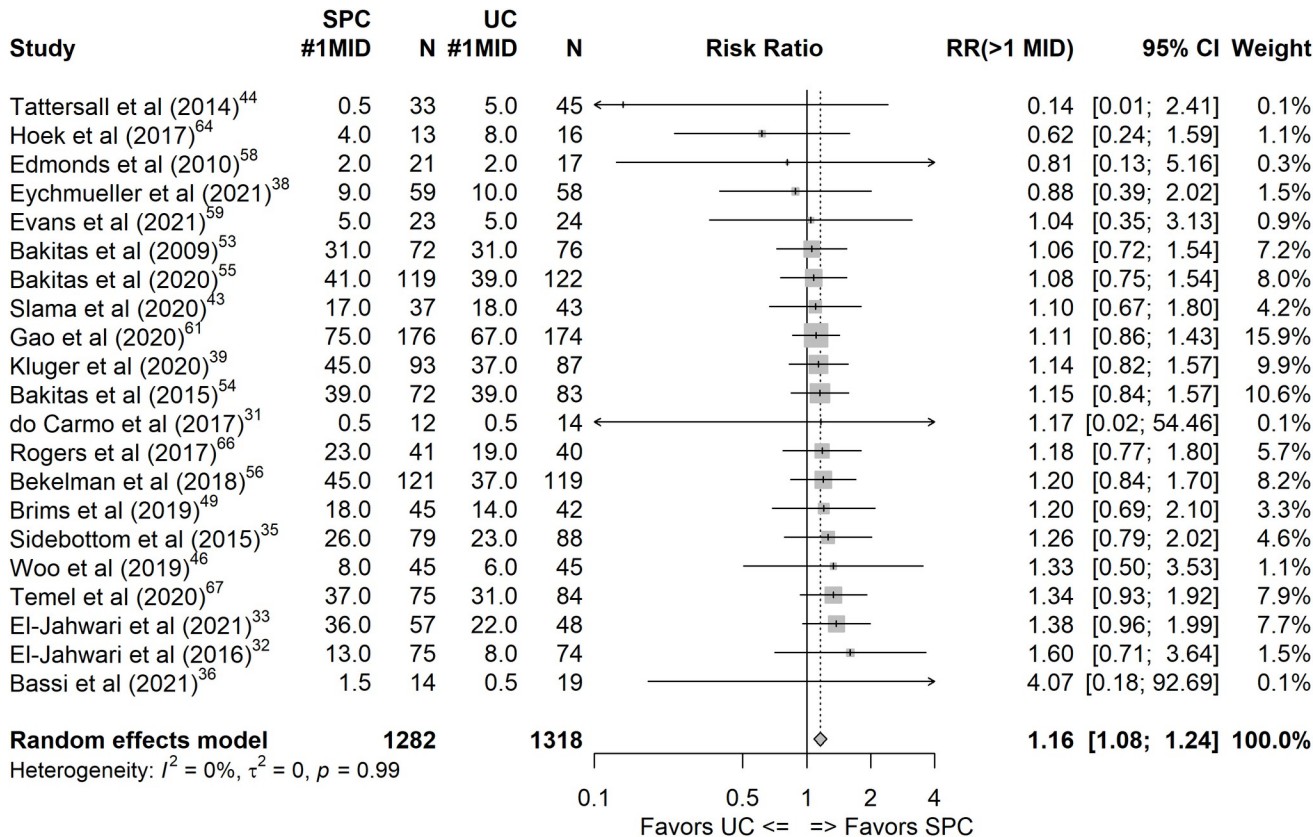

**Fig 3. Forest plot of the 13 to 36 months endpoint for the emotional wellbeing outcome, effect size: relative risk of experiencing a change of ≥ 1 MID between baseline and the endpoint.** #1MID, number of participants in the respective group experiencing change of at least 1 minimal important difference; CI, confidence interval; MID, minimal important difference; N, total number of participants in group; RR, relative risk; SPC, specialist palliative care; UC, usual care.

[70]. For example, a UK national hospice study found a median of 48 days between referral and death; 53 days for cancer versus 27 days for non-cancer [71]. However, early (prognosis of ≥6 months; n = 2) SPC trials showed greater effect sizes than trials of study populations with shorter prognoses (n = 5) [72], especially for those with higher baseline symptom severity [73,74]. Patient benefit from early SPC persisted long-term with better QoL at end-of-life than those receiving SPC only closer to death [72].

Our data showed the relative risk of experiencing a clinically important benefit increased as the duration of trial follow-up increased, and the NNT reduced after at least 3 months' follow up. For QoL outcomes, this was particularly so for people with non-cancer diseases, perhaps because SPC services have historically worked with cancer services fostering greater general palliative care skills and awareness. Our finding that a non-cancer diagnosis was associated with a higher effect size contrasts with Gaertner and colleagues [11], but their meta-analyses were based on only 2 trials.

Previous work—despite the lack of robust conclusions—points to the need for cross-setting, multidisciplinary service models. To our knowledge, our work is the first designed to examine the impact of service model on benefit.

Several strengths and limitations applied to this study. As a distinct strength of our work, the—to our knowledge—novel methodology, combining MIDs across different outcomes, allowed us to include more RCTs than previous meta-analyses, by combining different outcomes, and enabled robust conclusions regarding clinical importance of our findings. We

presented a clinically understood measure (i.e., NNT) allowing easier comparison with other interventions. Our sensitivity analysis indicated that benefits from SPC may be seen almost immediately.

Various limitations still applied. Our findings were still imprecise, but we could conclude that the benefit experienced was clinically important; previous point estimates of studies not using SMDs for MIDs have wide confidence intervals, with clinically questionable benefit for the lower bounds. The issue of contamination in 62% of studies may have led to an underestimation of effect. Effect sizes may have been over or underestimated by relying on a mixture of anchor and distribution-based approaches and thus, MIDs not being derived in palliative care samples. Further, included RCTs of palliative care interventions mainly represented evidence from high- and middle-income countries in which palliative care is often integrated and funded through the main healthcare system. The proportion of non-white participants was specified in 46% of included samples. The proportion of non-white participants ranged from 1% to 100%, with the majority of studies including around 15% of non-white participants. Therefore, the generalizability of our findings is limited to middle- and high-income countries with a majority of white participants. However, we managed to include studies in predominantly older populations with many studies including non-cancer conditions as well.

None of the prior proposals of service models [3,5–9,28] provided a summary score capturing both team skill mixes and range of service provision. Our bespoke classification scheme lacks validation and due to variable reporting of interventions, we may have underestimated the score, but results have face validity and support previous narrative findings [3,11–17].

In terms of implications for policy and practice, SPC does what it purports to do [1], providing better QoL and better emotional wellbeing for people with advanced disease. While we confirmed "better late than never," the usual situation of late referrals needs to be urgently addressed. Service provision should be multidisciplinary and integrated across healthcare settings which has implications for resourcing. Despite the World Health Assembly resolution of 2014 [75], progress towards universal access remains slow or nonexistent; commissioners fail to prioritize palliative care services. We showed an NNT similar to other accepted interventions, such as cardiac rehabilitation (NNT 12) [76]. Our findings will inform policy and service funders about the best model of care, sufficiently resourced to enable the most timely and effective intervention components, and a team with a skill mix able to provide the range of components needed. To do less is to risk the expense of an ill-equipped and ineffective service.

Regarding research, we showed higher effect sizes were associated with higher-quality trials with longer follow-up and less attrition. Future trials of palliative care models should include those able to benefit over a longer timescale to enable better demonstration of optimal benefit, with less bias due to missing data [77]. This will reduce research waste, improve the evidence base, and give greater impetus to practice implementation. A future similar meta-analysis with even more trials would improve precision.

In conclusion, we showed that SPC had a moderate overall effect in improving QoL and emotional concerns of people with life-limiting illness, regardless of medical condition. The most effective models of SPC service provision were found to be multidisciplinary, multicomponent, and multi-setting. Currently, most SPC referrals occur within weeks of death; our data seriously challenge this practice as too late for optimal benefit. Timely involvement in response to relevant concerns at any point during an individual's illness should be the standard of care.

## Ethics committee approval

No individual participant data were included in this meta-analysis. Ethics committee approval does not apply.

## Supporting information

**S1 Appendix. Review of publications presenting a classification system of models of specialist palliative care.**
(DOCX)

**S2 Appendix. Overview of all relevant systematic reviews and meta-analyses of effectiveness of specialist palliative care on quality of life and emotional wellbeing.**
(DOCX)

**S3 Appendix. Search strategy for database search in MEDLINE.**
(DOCX)

**S4 Appendix. Protocol for systematic review/meta-analysis.**
(DOCX)

**S5 Appendix. Tables with MIDs per outcome measure.**
(DOCX)

**S6 Appendix. Excluded full-texts with reasons for exclusion.**
(DOCX)

**S7 Appendix. Description of included studies.**
(DOCX)

**S8 Appendix. GRADE table and risk of bias table.**
(DOCX)

**S9 Appendix. Detailed results for all meta-analysis and meta-regressions.**
(DOCX)

**S10 Appendix. Detailed results for all sensitivity analyses.**
(DOCX)

**S11 Appendix. Completed PRISMA checklist.**
(DOCX)

## Author Contributions

**Conceptualization:** Miriam J. Johnson, Sophie Pask, Fliss E. M. Murtagh, Christina Ramsenthaler.

**Data curation:** Miriam J. Johnson, Leah Rutterford, Anisha Sunny, Sophie Pask, Susanne de Wolf-Linder, Fliss E. M. Murtagh, Christina Ramsenthaler.

**Formal analysis:** Leah Rutterford, Anisha Sunny, Sophie Pask, Fliss E. M. Murtagh, Christina Ramsenthaler.

**Funding acquisition:** Fliss E. M. Murtagh.

**Investigation:** Leah Rutterford, Sophie Pask, Susanne de Wolf-Linder, Fliss E. M. Murtagh, Christina Ramsenthaler.

**Methodology:** Christina Ramsenthaler.

**Project administration:** Miriam J. Johnson, Sophie Pask, Fliss E. M. Murtagh.

**Resources:** Miriam J. Johnson, Sophie Pask, Fliss E. M. Murtagh.

**Software:** Christina Ramsenthaler.

**Supervision:** Miriam J. Johnson, Sophie Pask, Fliss E. M. Murtagh, Christina Ramsenthaler.

**Validation:** Miriam J. Johnson, Sophie Pask, Susanne de Wolf-Linder, Fliss E. M. Murtagh, Christina Ramsenthaler.

**Visualization:** Susanne de Wolf-Linder, Christina Ramsenthaler.

**Writing – original draft:** Miriam J. Johnson, Leah Rutterford, Anisha Sunny, Sophie Pask, Fliss E. M. Murtagh, Christina Ramsenthaler.

**Writing – review & editing:** Miriam J. Johnson, Leah Rutterford, Anisha Sunny, Sophie Pask, Susanne de Wolf-Linder, Fliss E. M. Murtagh, Christina Ramsenthaler.

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
