## [Editor Report · Decision Letter 0]

13 Mar 2024

Dear Dr Ramsenthaler, 

Thank you for submitting your manuscript entitled "Does specialist palliative care provide additional benefits? A meta-analysis with meta-regression to identify active ingredients of service composition, service structure, and delivery model" for consideration by PLOS Medicine.

Your manuscript has now been evaluated by the PLOS Medicine editorial staff and I am writing to let you know that we would like to send your submission out for external peer review.

Please re-submit your manuscript within two working days, i.e. by Mar 15 2024.

Feel free to email me at aschaefer@plos.org or us at plosmedicine@plos.org if you have any queries relating to your submission.

Kind regards,

Alexandra Schaefer, PhD

Associate Editor

PLOS Medicine

---

## [Decision Letter · Decision Letter 1]

12 Apr 2024

Dear Dr. Ramsenthaler,

Thank you very much for submitting your manuscript "Does specialist palliative care provide additional benefits? A meta-analysis with meta-regression to identify active ingredients of service composition, service structure, and delivery model" (PMEDICINE-D-24-00814R1) for consideration at PLOS Medicine. 

Your paper was evaluated by an associate editor and discussed among all the editors here. It was also discussed with an academic editor with relevant expertise, and sent to independent reviewers, including a statistical reviewer. The reviews are appended at the bottom of this email and any accompanying reviewer attachments can be seen via the link below:

[LINK]

In light of these reviews, I am afraid that we will not be able to accept the manuscript for publication in the journal in its current form, but we would like to consider a revised version that addresses the reviewers' and editors' comments. Obviously we cannot make any decision about publication until we have seen the revised manuscript and your response, and we plan to seek re-review by one or more of the reviewers. 

Please use the following link to submit the revised manuscript: https://www.editorialmanager.com/pmedicine/

We expect to receive your revised manuscript by May 03 2024. However, if this deadline is not feasible, please contact me by email, and we can discuss a suitable alternative.

Don't hesitate to contact me directly with any questions (aschaefer@plos.org). If you reply directly to this message, please be sure to 'Reply All' so your message comes directly to my inbox.

We look forward to receiving your revised manuscript.

Sincerely,

Alexandra Schaefer, PhD

PLOS Medicine

plosmedicine.org

ACADEMIC EDITOR COMMENTS

I agree that it's an interesting approach to an important topic. The areas that I don't think are covered are the context of who is dying where around the world of what, and then contextualizing the results in the discussion with that in mind. It's possible that the trials have been done effectively in populations where there's almost a pre-selection of potential to benefit from palliative care services - and that's where it's important to look at settings, age groups, conditions that people are dying from. The countries and regions of the world are not provided, and it seems likely that most of this evidence is from high-income countries with a handful represented, even before thinking about the different settings in which people die. Caution must therefore be exercised in interpretation and generalization.

EDITORIAL COMMENTS

The Editorial team agrees that your study addresses an important topic, but we feel that the study could be more specific about how the study (perhaps including the study design) adds to existing knowledge and, for example, how it highlights a gap in current service provision. Please consider our input as you revise your manuscript.

***Please note: not all will apply to your paper, but please check each item carefully

GENERAL COMMENTS

1) Please cite the reference numbers in square brackets. Citations should be preceding punctuation.

COMPETING INTEREST

All authors must declare their relevant competing interests per the PLOS policy, which can be seen here: https://journals.plos.org/plosmedicine/s/competing-interests

For authors with ties to industry, please indicate whether any of the interests has a financial stake in the results of the current study.

ABSTRACT

1) Please report your abstract according to PRISMA for Abstracts, following the PLOS Medicine abstract structure (Background, Methods and Findings, Conclusions).

2) PLOS Medicine requests that main results are quantified with 95% CIs as well as p values. When reporting p values please report as p<0.001 and where higher as the exact p value p=0.002, for example. For the purposes of transparent data reporting, if not including the aforementioned please clearly state the reasons why not. When a p value is given, please specify the statistical test used to determine it.

3) Throughout, suggest reporting statistical information as follows to improve clarity for the reader “22% (95% CI [13%,28%]; p</=)”. Please be sure to define all numerical values at first use. Please amend throughout the abstract and main manuscript. Please note the use of commas to separate upper and lower bounds, as opposed to hyphens as these can be confused with reporting of negative values.

4) Please ensure that all numbers presented in the abstract are present and identical to numbers presented in the main manuscript text.

5) Please provide the dates of search, data sources, number of studies included, types of study designs included, eligibility criteria, and synthesis/appraisal methods. 

6) Please define all abbreviations including those for statistical reporting at first use.

7) In the last sentence of the Abstract Methods and Findings section, please describe the main limitation(s) of the study's methodology.

AUTHOR SUMMARY

At this stage, we ask that you include a short, non-technical Author Summary of your research to make findings accessible to a wide audience that includes both scientists and non-scientists. The Author Summary should immediately follow the Abstract in your revised manuscript. This text is subject to editorial change and should be distinct from the scientific abstract. Ideally each sub-heading should contain 2-3 single sentence, concise bullet points containing the most salient points from your study. In the final bullet point of ‘What Do These Findings Mean?’, please include the main limitations of the study in non-technical language. Please see our author guidelines for more information: https://journals.plos.org/plosmedicine/s/revising-your-manuscript#loc-author-summary

INTRODUCTION

Please address past research and explain the need for and potential importance of your study. Indicate whether your study is novel and how you determined that. If there has been a systematic review of the evidence related to your study (or you have conducted one), please refer to and reference that review and indicate whether it supports the need for your study. 

METHODS AND RESULTS

1) PLOS Medicine requests that main results are quantified with 95% CIs as well as p values. We suggest reporting statistical information as detailed above – see under ABSTRACT

2) Please present numerators and denominators for percentages (at least in the Tables [not necessarily each time they're mentioned]).

3) Please report your study according to the PRISMA guidelines provided at the EQUATOR site. Please add the following statement, or similar, to the Methods: "This study is reported as per the Preferred Reporting Items for Systematic Reviews and Meta-Analyses (PRISMA) guideline (S1 Checklist)."

4) Please provide the completed PRISMA checklist as Supporting Information. When completing the checklist, please use section and paragraph numbers, rather than page numbers.

DISCUSSION

Please present and organize the Discussion as follows: a short, clear summary of the article's findings; what the study adds to existing research and where and why the results may differ from previous research; strengths and limitations of the study; implications and next steps for research, clinical practice, and/or public policy; one-paragraph conclusion (no subheading).

FIGURES AND TABLES 

1) Please provide titles and legends for all figures and tables (including those in Supporting Information files). 

2) Please define all abbreviations used in each figure/table (including those in Supporting Information files). 

3) Please consider avoiding the use of red and green in order to make your figure more accessible to those with color blindness. 

SUPPLEMENTARY MATERIAL

1) For supplementary figures and tables, please see the general comments under TABLES and FIGURES and amend accordingly.

2) We suggest reporting statistical information as detailed above – see under ABSTRACT. Please be sure to define all numerical values.

3) As for the main manuscript, please indicate whether analyses are adjusted to help facilitate transparent data reporting please also detail the factors adjusted for and present the unadjusted analyses for comparison. If not, please clearly state the reasons why not.

4) Please cite your Supporting Information as outlined here: https://journals.plos.org/plosmedicine/s/supporting-information

REFERENCES

1) PLOS uses the numbered citation (citation-sequence) method and first six authors, et al.

2) Please ensure that journal name abbreviations match those found in the National Center for Biotechnology Information (NCBI) databases (http://www.ncbi.nlm.nih.gov/nlmcatalog/journals), and are appropriately formatted and capitalised.

3) Where website addresses are cited, please specify the date of access (e.g. [accessed: 16/09/2023]).

4) Please also see https://journals.plos.org/plosmedicine/s/submission-guidelines#loc-references for further details on reference formatting. 

Comments from the reviewers:

Reviewer #1: This is a novel approach to addressing the challenges of doing meta-analyses of widely heterogenous studies and is important to do in order to demonstrate the benefits of early integrated specialist palliative care, with multidisciplinary components. The results are clinically meaningful and relevant in these times of fiscal scarcity and fierce competition for resources. 

My review is hampered by my lack of statistical expertise and I would recommend that it be thoroughly reviewed by someone with such expertise before accepting my recommendation. 

From the clinical perspective and being knowledgeable about the topic I think it is well-written, makes sense, and I think important to be published. 

I have no suggestions for improvement from a grammatical or language perspective. 

Reviewer #2: Firstly, I would like to thank the authors for their work in putting this manuscript together. I appreciate the enormity of the task that has been undertaken. The authors aimed to assess the impact of special palliative care on patient reported outcomes of quality of life and emotional wellbeing. I have mainly focused on the statistical analysis. Below are some queries: 

Methods: The main text doesn't appear to mention distribution based methods at all, only anchor based (page 8, line 124).

Search date restriction: Could the authors please provide reasoning for this restriction and the impact it may have had on the identification of other studies of relevance before this date.

Study selection: Information on how studies were screened is lacking information, e.g. what software was used. I would also suggest including study selection in the header "eligibility" as this will aid readability of the manuscript. 

DSL method is used and given that the majority of analyses include 20+ studies, this is probably OK. However, I would recommend the authors use a Hartung-Knapp correction to assess the impact on the analyses. Additionally, as some analyses include <20 studies, I feel this correction should be applied in those analyses. I would recommend a sensitivity analysis conducting both methods and using the most conservative of the two methods (i.e. the method with the widest confidence intervals) for each analysis. 

Forest plots: where there is heterogeneity, please include prediction intervals. 

Anchor based MID: What were the thresholds considered to qualify the MID? Did the authors consider the correlation between the anchor and the outcome? These details should be added to the report for the assessment of the validity of the anchors. 

Distribution based MID: what method was used to calculate the distribution based MID? There is detail in the supplementary material regarding the calculation of the SMD but no information appears to be present for method of calculating MID. What SD did the authors use for the calculation of the MID? Could the authors please provide this information. 

Subgroup analysis: While the authors have conducted meta-regression, I would like to ask if they had/have considered using subgroup analyses? I think it would be important to understand the direction of effects in some of the subgroups. For example, cancer/non-cancer/mixed. Did the authors consider and run these analyses? I think it would show some beneficial information and provide further clarification as to whether there are any differences between these subgroups in terms of quality of life and emotional well-being.

Table 2: I think you should include a measure of heterogeneity (preferably tau) in this table as the results alone could be misleading. I appreciate these are in the supplementary information but I feel there is a need to be clear about the level of statistical heterogeneity in the main text as well. 

Reviewer #3: Dear Authors,

The topic is really interesting, important and useful for the health sector, but to be published, it needs important advice:

TITLE:

The title needs to be better rewritten, avoiding the repetition of the word "service". The title needs to show the content of the article.

SUMMARY:

Separate the items Methods and Results. In Methods, specify the type of study carried out and improve the writing, which is a little confusing.

In Results, we summarize the main results of the study.

Summary Conclusions:

It is necessary to respond to the objectives of the study and be in tune with the Conclusions presented at the end of the study.

INTRODUCTION;

Please review the text to make it clearer, less confusing and specific to the title.

Make clear in the Introduction, the Question (question) and the Importance of the research

METHODS:

Explain the Type of Search. Check whether all recommendations on the PRISMA 2020 list have been met.

In Line 106, Data extraction and vision risk assessment, the authors say that a data deletion spreadsheet was developed and tested. I asked: was it created and tested where, how and by whom? please quote.

Line 111, Please improve the wording of the sentence related to the risk of bias.

All text regarding Data Analysis and Precise Review Results. It's confusing.

CONCLUSION:

In conclusion, it is necessary to respond to the objectives, which are: Estimate the summary effect of specialized palliative care in all environments on quality of life and emotional well-being and identify the ideal model of service provision. The Conclusion that is in the SUMMARY must be, in short, the same that is here at the end.

[LINK]

General journal requests:

1) Please ensure that the paper adheres to the PLOS Data Availability Policy (see http://journals.plos.org/plosmedicine/s/data-availability), which requires that all data underlying the study's findings be provided in a repository or as Supporting Information. For data residing with a third party, autho

---

## [Decision Letter · Decision Letter 2]

31 May 2024

Dear Dr. Ramsenthaler,

Thank you very much for re-submitting your manuscript "Does specialist palliative care provide additional benefits? A meta-analysis with meta-regression to identify active ingredients of service composition, service structure, and delivery model" (PMEDICINE-D-24-00814R2) for review by PLOS Medicine.

Thank you for your detailed response to the editors' and reviewers' comments. I have discussed the paper with my colleagues and the academic editor, and it has also been seen again by the two of the original reviewers. The changes made to the paper were satisfactory to the reviewers. As such, we intend to accept the paper for publication, pending your attention to the editorial comments below in a further revision. When submitting your revised paper, please once again include a detailed point-by-point response to the editorial comments.

[LINK]

In revising the manuscript for further consideration here, please ensure you address the specific points made by each reviewer and the editors. In your rebuttal letter you should indicate your response to the reviewers' and editors' comments and the changes you have made in the manuscript. Please submit a clean version of the paper as the main article file. A version with changes marked must also be uploaded as a marked up manuscript file. Please also check the guidelines for revised papers at http://journals.plos.org/plosmedicine/s/revising-your-manuscript for any that apply to your paper. 

We ask that you submit your revision within 1 week (Jun 07 2024). However, if this deadline is not feasible, please contact me by email, and we can discuss a suitable alternative.

Please do not hesitate to contact me directly with any questions (atosun@plos.org). If you reply directly to this message, please be sure to 'Reply All' so your message comes directly to my inbox.

We look forward to receiving the revised manuscript.   

Sincerely,

Alexandra Tosun, PhD

Associate Editor 

PLOS Medicine

plosmedicine.org

Requests from Editors:

Please update the Financial Disclosure Statement on the online submission form with the information provided on lines 408-413.

ABSTRACT

1) l.33: Please change to: “We conducted a..”.

2) l.39: Please temper claims of primacy of results by stating, "to our knowledge" or something similar (i.e. “The meta-analysis used, to our knowledge, novel methodology to convert outcomes into minimally clinically…”).

3) ll.45-46: “almost exclusively from high and middle-income countries” – Thank you for including this detail. We suggest including an exact number.

4) l.47: Please delete ‘with’ before the brackets.

5) l.48: Please define “CI” at first use.

6) l.48: “effect size of 0.40 at 4 to 6 months”- in the table the time span is “at 13 to 36 weeks”, please revise and make sure this is consistently reported throughout.

7) ll.49-50: For clarity, please re-iterate the time frame this refers to.

8) l.50: Please define “RR” at first use in the Abstract.

9) l.50/l.51: In line 50, you mention "1 MID magnitude change" and provide a risk ratio, whereas in the following line you describe the results as "1 MID unit change" and the corresponding risk ratio is missing (RR 1.16; l.298). Please revise. We have also noted that in the abstract you refer to the time frame as "4 to 6 months", whereas in the main text and Table 2 this time frame is referred to as "13 to 36 weeks" (which is not the same as 4 to 6 months) - please revise and use a consistent format (see comments above).

10) l.56: “lower number-needed-to-treat (14)” – We feel the number 14 here is rather confusing without appropriate context. We suggest removing it.

11) l.61: Please temper claims of primacy of results by stating, "to our knowledge" or something similar.

AUTHOR SUMMARY

Thank you for providing the Author Summary. The Author Summary in its current format is rather long and needs to be shortened. It should contain only the most important details for each of the questions, in a condensed form. For example, the bullet points of “Why was this study done?”, could be condensed as follows (please note that the last bullet point belongs under the second question “What did the researchers do and find?”): 

- Specialist palliative care (SPC) services provide a complex intervention that addresses the holistic needs of individuals with life-limiting conditions and their families.

- Different intervention models include a variation of different disciplines (doctors, nurses, psychologists, etc.), configurations (e.g., whether out-of-hours care is provided), and settings (hospital, hospice, community, etc.)

- The overall effectiveness of SPC on quality of life and emotional wellbeing is undetermined due to large variation in intervention models and heterogeneity in outcome measures used to measure quality of life and emotional wellbeing. 

- We systematically reviewed evidence for the effectiveness of SPC, to assess which intervention model components and configurations are most effective in improving quality of life or emotional wellbeing.

It may be helpful to review currently published articles for examples which can be found on our website here https://journals.plos.org/plosmedicine/

INTRODUCTION

l.160: Please temper claims of primacy of results by stating, "to our knowledge" or something similar.

METHODS AND RESULTS

1) In the Methods section, we suggest adding a brief explanation on the calculation of the number needed to treat (with reference to the Supplementary Appendix), together with the interpretation of the NNT.

2) ll.172/173: We think it might be helpful to explain the data restriction choice in more detail, as you did in response to the reviewer comments. Please be sure to include a reference. For example: "Given that SPC service models have become part of mainstream healthcare only in recent years, and that for most countries in Europe or the United States, service models and their components have been developed since the 2000s, we restricted the search to synthesizing effects for contemporary service models.”

3) l.249: “RCTs by Liu et al [28] and Nottelmann et al [39] included the most diverse group.” – in terms of different professions involved? Please clarify.

4) l.257: Please change to “populations with cancer”. We prefer the use of patient-centered language. Please revise accordingly throughout the main text. 

5) l.258: “Of these, 19 RCTs used mixed cancer populations.” – Please clarify what mixed cancer populations means (different types of cancer?).

6) ll.259-261: “The mean age was 65.8 years...” – if possible, please provide standard deviation and interquartile ranges when presenting mean and median values.

7) l.271: Please define “UC” at first use.

8) l.275: Please define “NNT” at first use.

9) l.283: “The covariate consistently associated with the MD(MID) was the attrition..” – should this be “SMD”?

10) l.289: Please check whether you meant “SMD” instead of “MD”.

11) l.290: Please change to “(p=0.003)”.

FIGURES AND TABLES

1) Figure 1: Please define “RCT” in the figure description or write in full in the figure.

2) Table 1: Please define “SPC” and “QoL” at first use.

3) Table 3: Please either define “SPC” in the title and “ACP” in the table or list both definition below the table.

DISCUSSION

1) Please remove any subheadings from the Discussion.

2) ll.316-319: “The NNT for one patient to have benefit of at least 1 MID improves markedly for both types of measures in studies with follow-up of between 3 and 6 months; QoL outcomes improve from 74 (<3 months) to 16 (4 to 6+ months), and emotional outcomes improve from 49 (<3 months) to 24 (4 to 6 months).” – in the second part of the statement, please reiterate that the numbers you are describing are the numbers needed to treat. In addition, we have noticed that the numbers presented here do not match the numbers presented in the abstract (ll.52-52). We are not sure where the numbers come from, please check and comment.

3) l.350: Please temper claims of primacy of results by stating, "to our knowledge" (or something similar), or removing the word “novel”.

4) ll.350-352, please change to: “As a distinct strength of our work, our methodology, combining MIDs across different outcomes, allowed us to include more RCTs than previous meta-analyses and enabled robust conclusions regarding clinical importance of our findings.”

SUPPLEMENTARY MATERIAL

Thank you for providing the completed PRISMA checklist. We ask you to revise the checklist using section and paragraph numbers, when completing the checklist (i.e. please specify “Please see paragraph in manuscript”). 

SOCIAL MEDIA

To help us extend the reach of your research, please provide any X (formerly known as Twitter) handle(s) that would be appropriate to tag, including your own, your co-authors’, your institution, funder, or lab. Please enter in the submission form any handles you wish to be included when we post about this paper.

Comments from Reviewers:

Reviewer #2: The authors have provided detailed and well written responses. Where they could accommodate the reviewers they have, when they could not, they provide well justified arguments and clear reasoning. From a statistical standpoint they have conducted the meta-analysis using the best available data. Again, I would just like to show my appreciation to the work that has gone into this review. 

Reviewer #3: The topic developed is interesting and useful for health professionals. In the previous review, I sent important comments and suggestions to the authors of the article. In the current review, I verify that the suggestions were considered and that the article was clear and easy for the reader to understand. The title is better and represents the content covered in the article. The details regarding the methods allow reproduction by other researchers. The conclusions respond to the objective of the article.

[LINK]

General Editorial Requests

---

## [Editor Report · Decision Letter 3]

21 Jun 2024

Dear Dr. Ramsenthaler,

Thank you very much for re-submitting your manuscript "Does specialist palliative care provide additional benefits? A meta-analysis with meta-regression to identify active ingredients of service composition, structure, and delivery model" (PMEDICINE-D-24-00814R3) for review by PLOS Medicine.

I have discussed the paper with my colleagues and there remain a few outstanding requests which need to be addressed prior to publication.

[LINK]

We ask that you submit your revision within 1 week (Jun 28 2024). However, if this deadline is not feasible, please contact me by email, and we can discuss a suitable alternative.

Please do not hesitate to contact me directly with any questions (atosun@plos.org). If you reply directly to this message, please be sure to 'Reply All' so your message comes directly to my inbox.

We look forward to receiving the revised manuscript.

Sincerely,

Alexandra Tosun, PhD

Associate Editor 

PLOS Medicine

plosmedicine.org

Requests from Editors:

1) Please revise your title according to PLOS Medicine's style. Your title must be nondeclarative and not a question. It should begin with the main concept if possible. "Effect of" should be used only if causality can be inferred, i.e., for an RCT. Please place the study design ("A randomized controlled trial," "A retrospective study," "A modelling study," etc.) in the subtitle (i.e., after a colon). For example: Benefits of specialist palliative care by identifying active ingredients of service composition, structure, and delivery model: A systematic review with meta-analysis and meta-regression

2) Abstract: As you are reporting results for NNT in the Abstract, please mention this in the Methods section of the Abstract. For example: "We also calculated the number needed to treat (NNT)."

3) l.47: “…benefit was seen from four months’ follow-up..” – should this say three months, referring to the 13 (to 36) weeks? Please check and revise.

4) ll.61-64, please change to: “Using, to our knowledge, novel methods to combine different outcomes, we show clear evidence of moderate overall effect size for both quality-of-life and emotional wellbeing benefits from SPC, regardless of underlying condition, with multi-disciplinary, multi-component, and multi-setting models being most effective.”

5) l.86: Please remove the word “all”.

6) l.105: Please create a new bullet point starting with “Honoring the complex and…”.

7) l.284: “at our primary endpoint of 13 to 38 weeks” – please change to ’36 weeks’. 

8) l.302ff: Please revise the Discussion for tense. Results and findings should be discussed in the past tense, e.g. “For QoL and emotional outcomes, statistically and clinically significant benefits were seen in trials with follow up for three months or more.”.

9) ll.304-307, please change to: “The NNT for one patient to have benefit of at least 1 MID improved for both types of measures in studies with follow-up of between 3 and 6 months. The NNT for QoL outcomes improved from 69 (<3 months) to 15 (12 weeks) and 20 (13+ weeks), and the NNT for emotional outcomes improved from 46 (<3 months) to 19 (13 to 36 weeks).”

10) Figure 1: You have included the definition of 'RCT' in the list of abbreviations for Figure 2 instead of Figure 1. Please revise (you may write 'RCT' in full in the flow diagram).

Comments from Reviewers:

[LINK]

General Editorial Requests

---

## [Editor Report · Decision Letter 4]

28 Jun 2024

Dear Dr Ramsenthaler, 

On behalf of my colleagues and the Academic Editor, Carol Brayne, I am pleased to inform you that we have agreed to publish your manuscript "Benefits of specialist palliative care by identifying active ingredients of service composition, structure, and delivery model: A systematic review with meta-analysis and meta-regression" (PMEDICINE-D-24-00814R4) in PLOS Medicine.

I appreciate your thorough responses to the reviewers' and editors' comments throughout the editorial process. We look forward to publishing your manuscript, and editorially there are only a few remaining minor stylistic/presentation points that should be addressed prior to publication. Please go over the discussion carefully one more time for the use of the appropriate tense. We will carefully check whether the changes have been made. If you have any questions or concerns regarding these final requests, please feel free to contact me at atosun@plos.org.

Please see below the minor points that we request you respond to:

1) l.61, please change to: “…we found clear evidence…”

2) l.328, please change to: “Our data showed…”

3) l.340, please change to: “We presented…”

4) l.371, please change to: “Regarding research, we showed…”

5) Please remove the "Conclusions" subheading. The discussion should be a continuous text. We suggest changing the opening statement of the conclusion paragraph to: "In conclusion, we showed that SPC had a moderate overall effect in improving QoL and emotional concerns of people with life-limiting illness, regardless of medical condition."

6) l.381, please change to: “The most effective models of SPC service provision were found to be multidisciplinary, multi-component, and multi-setting.”

7) Figure 2: Please remove the definition for "RCT" from the list of abbreviations.

PRESS

Sincerely, 

Alexandra Tosun, PhD 

Associate Editor 

PLOS Medicine